# Assessing Physical Activity Levels among Chinese College Students by BMI, HR, and Multi-Sensor Activity Monitors

**DOI:** 10.3390/ijerph20065184

**Published:** 2023-03-15

**Authors:** Dansong Liu, Xiaojuan Li, Qi Han, Bo Zhang, Xin Wei, Shuang Li, Xuemei Sui, Qirong Wang

**Affiliations:** 1School of Physical Education, Hubei University of Technology, Wuhan 430068, China; 2Sports Nutrition Center, National Institute of Sports Medicine, Beijing 100029, China; 3School of Physical Education, Hubei University, Wuhan 430062, China; 4College of Physical Education, Guangzhou University, Guangzhou 510006, China; 5Department of Exercise Science, Arnold School of Public Health, University of South Carolina, Columbia, SC 29208, USA

**Keywords:** energy expenditure monitoring, algorithm model, physical activity monitors

## Abstract

We investigated the use of multi-sensor physical activity monitors, body mass index (BMI), and heart rate (HR) to measure energy expenditure (EE) of various physical activity levels among Chinese collegiate students, compared with portable indirect calorimetry. Methods: In a laboratory experiment, 100 college students, 18–25 years old, wore the SenseWear Pro3 Armband™ (SWA; BodyMedia, Inc., Pittsburg, PA, USA) and performed 7 different physical activities. EE was measured by indirect calorimetry, while body motion and accelerations were measured with an SWA accelerometer. Special attention was paid to the analysis of unidirectional and three-directional accelerometer output. Results: Seven physical activities were recorded and distinguished by SWA, and different physical activities demonstrated different data features. The mean values of acceleration ACz (longitudinal accel point, axis Z) and VM (vector magnitude) were significantly different (*p* = 0.000, *p* < 0.05) for different physical activities, whereas no significant difference was found in one single physical activity with varied speeds (*p* = 0.9486, *p* > 0.05). When all physical activities were included in a correlation regression analysis, a strong linear correlation between the EE and accelerometer reporting value was found. According to the correlation analysis, sex, BMI, HR, ACz, and VM were independent variables, and the EE algorithm model demonstrated a high correlation coefficient R^2^ value of 0.7. Conclusions: The predictive energy consumption model of physical activity based on multi-sensor physical activity monitors, BMI, and HR demonstrated high accuracy and can be applied to daily physical activity monitoring among Chinese collegiate students.

## 1. Introduction

The need for reliable methods to accurately calculate daily energy expenditure is important for public health supervision, especially for monitoring the epidemic of overweight and obesity among various populations. Physical activity is associated with a reduced risk of morbidity and mortality in many chronic diseases, including CVD, diabetes, obesity, and some tumors [1,2]. Physical activities have also been shown to promote body weight control with appropriate nutrition and energy intake. The ability to accurately assess energy expenditure (EE) in free-living individuals can enhance the knowledge related to the link between the dose of physical activity and health status, as well as improve the understanding of how energy expenditure impacts energy balance related to body weight control and chronic diseases such as diabetes mellitus [3]. Research on the energy expenditure of physical activity can identify the recommended amount of targeted daily physical activity, which can be specific and personalized, such as the recommended amount of physical activity for different ages, genders, and population cohorts (e.g., for obese and overweight people). In addition to these benefits, a significant number of residents in China do not participate in sufficient amounts of physical activity to achieve its health benefits.

A large sample size is always preferred in a physical activity and health study. Traditional physical activity research is majorly based on questionnaires, which is less objective. One of the challenges in energy expenditure research is the ability to accurately assess physical activity in free-living individuals. Numerous methods are available, but each of the current methods to assess physical activity and energy expenditure has limitations [4]. In the last decades, scientists have developed direct calorimetry (DC), doubly labeled water (DLW), indirect calorimetry (IC), and motion sensors for measuring energy expenditure with different physical activities (PAEEs), from which, DC, IC, and DLW are considered to be the gold standard of measuring [5]. Meanwhile, direct measuring is expensive and restricts people to a small and limited space, and it is not feasible to conduct a large cohort study at the same time. Doubly labeled water (DLW) is considered to be the most accurate technique to assess energy expenditure. The DLW method calculates the energy consumption of the human body according to the elimination rate of ^2^H and ^18^O from the human body within a period of time (5–14 days). It has the advantages of high accuracy, convenient sampling, wide application, and long-term monitoring, but it is expensive and cannot measure energy consumption in a short time for specific physical activity; therefore, its application is limited [6]. The IC method is also known as the gas metabolism method. Its principle is to estimate energy consumption by measuring gas exchange in human breath. Its main instruments are represented by COSMED K4 series and CORTEX METAMAX3B. However, these devices can only carry out monitoring for a short period of time (such as 2–4 h) and need to be equipped with a computer terminal near the test site. The subjects are also prone to discomfort when wearing breathing masks, so the IC method is limited to the energy consumption test under the condition of small sample sizes and short-term exercise. Self-reporting physical activity questionnaires have the limitation of reporting accuracy, compared with objective techniques. Motion sensors mainly include accelerometers and pedometers, which can indirectly reflect the level of physical activity by recording the vibration of the body during physical activity. A motion sensor is easy to wear and has high accuracy, so it has a wide application prospect in physical activity monitoring. However, motion sensors cannot accurately monitor energy consumption in activities, such as cycling and static fitness, due to small body vibration, which is a major limitation of motion sensors [7]. Because of the cost and technical demands, these methods are limited to small, pilot, and validation studies. Methods with good validity tend to be very costly or complicated for outdoor activities; however, practical and feasible methods for large populations have the limitation of poor accuracy and/or reliability.

Developing an objective and accurate method to assess energy expenditure associated with physical activity continues to be a need in the field. Accelerometers are commonly used because they are practical and effective. Many commercial accelerometers are currently available, and there is considerable confusion over the appropriate test index and method to convert accelerometer counts into estimates of physical activity or energy expenditure [8,9]. As the main tool of this physical activity research, most accelerometer data analysis methods and energy consumption prediction models are based on the daily physical activity of adults. Because of the different characteristics of accelerometers, determining the accuracy and efficiency of accelerometers has become the key factor in relevant research. CSA, RT3, and SWA accelerometers have been widely used in research in recent years, and a large number of experiments have been carried out to illustrate their accuracy and applicability for energy consumption prediction [10,11]. Their methods and experimental designs are advanced and comprehensive.

With the increasing understanding of the correlation between physical activity and health, the monitoring of physical activity EE and exercise intervention methods have gradually become a key focus of physical activity research. Scholars have been seeking a convenient, accurate, and practical monitoring method to effectively calculate and evaluate the energy consumption generated by physical activity. This method should first identify the differences in physical activities, and then calculate energy consumption by an accurate algorithm method. Therefore, this study, using a population of Chinese college students, establishes an energy consumption model by monitoring seven kinds of physical activities with a triaxial acceleration sensor. Through the calculation and analysis of the data available, a graph is drawn to verify the triaxial acceleration sensor in distinguishing different types of body activity, and the accuracy of the energy consumption algorithm is improved by adding additional relevant parameters (BMI, HR, and sex).

## 2. Materials and Methods

### 2.1. Participants

One hundred Chinese collegiate students (aged 18–25) were recruited to participate in this research (as shown in Table 1). Subjects who met the exclusion criteria were not included (exclusion criteria: major disease or illness, use of medications that would affect body weight or metabolism, a current smoker). Participants were aware of the procedures and purposes of the study before they signed the informed consent documents. Participants were required to eat normally one week before the test and not to eat within 2 h before testing. A total of 80 participants enrolled as EXPERIMENTAL GROUP and took part in the test, and the other 20 enrolled as CONTROL GROUP to verify the accuracy of the model.

### 2.2. Measurement of Descriptive Variables

All participants were in light-weight clothing (shorts, t-shirt, and barefoot) before the exercise sessions. Each subject arrived at the laboratory one hour before the test started. Weight and height were determined without shoes. Body weight was measured to the nearest 0.1 kg using a JY-200D height/weight digital scale (Jingyi equipment Co., Ltd., Beijing, China), and body height was determined to the nearest 0.1 cm using a horizontal headboard with an attached wall-mounted metric rule. Body mass index (BMI, kg/m^2^) was computed from weight and height.

### 2.3. Exercise Protocols

Seven separate activities were included in this project (rest, sit, treadmill walk, normal run, stair walk, cycle ergometer, push-ups). The specific protocols are shown in Table 2. The order of these exercise protocols was performed randomly. Walking was performed on a motorized treadmill. Stair walk was performed with a 20.3 cm (6-inch) bench stair in the following sequence: left leg up, right leg up, left leg down, right leg down. Cycle ergometer exercise was performed on a Monark 827E stationary cycle ergometer (Monark Exercise AB, Vansbro, Sweden). A metronome was used to set pace during stair stepping and performing push-ups. During each activity, energy expenditure was measured simultaneously. Exercise heart rate was used to assess exercise intensity at each minute using a Polar heart rate monitor [12,13].

The experiments were carried out in the Exercise Physiology Laboratory (Wuhan Sports University, Wuhan, China), and the room was maintained at a temperature of 20~25 °C and 50~80% humidity. Participants were asked to perform the test one after another. In order to strictly control the airflow, other subjects and staff were not allowed to enter the room without permission during the test. The research project was approved by the Bioethics Committee of the Academy of Physical Education of Wuhan Sports University (Wuhan, China) and was conducted in accordance with the Declaration of Helsinki. The participants were informed and fully aware of the procedures of this study and acknowledged all the risks and benefits before they were recruited. All participants conducted the physical activities as they were required (as shown in Table 2).

### 2.4. Motion Sensors

The SenseWear Pro3 Armband™ (Body Media, Pittsburgh, PA, USA) is a commercially available, comfortable, non-invasive physical activity monitor that is worn on the upper arm over the triceps muscle and provides information about body position (lying or upright) by detecting accelerations. The armband was placed on each subject’s arm before entering the laboratory, and the subject remained in a seated position for a period of 5 min before data collection to allow for acclimation to skin temperature. Energy expenditure during exercise was captured at 60 s intervals, the real-time sampling frequency was checked every 0.1 s, and the peak acceleration and mean absolute difference (MAD) values were captured every 6 s.

### 2.5. Energy Expenditure

Energy expenditure during various exercises was examined via open-circuit, indirect calorimetry with the Cortex MetaMax3B^TM^ metabolic system (Cortex Biophysik GmbH, Leipzig, Germany). Cortex MetaMax3B^TM^ system, which is a battery-operated, portable, wireless metabolic system measuring gas exchange breath-by-breath, has been reported to be a valid and reliable measure of oxygen uptake [12]. The face mask was connected to a flow sensor to detect airflow from the rotation of fans, which allowed the determination of ventilation. A sampling line connected both flow sensor unit and sensor box; oxygen (O_2_) and Carbon Dioxide (CO_2_) from expired air were analyzed using a micro-fuel cell and thermal conductivity, respectively. Cortex MetaMax3B software was used to compute energy expenditure, which included oxygen uptake in milliliters per minute (mL/min), milliliters per kilogram of body weight per minute (mL/kg/min), and kilocalories per minute (Kcal/min). Experiments were carried out in a regular temperature and humidity lab setting for all exercise tests, airflow calibration was performed using an automatic flow calibrator, and the gas analyzers were calibrated (5% O_2_, 16% CO_2_, and 79% N_2_). All experiments were performed for at least 5 min, and a minimum of 3 min of resting gas exchange assessment data were collected both before and after each of the tests. The purpose of the resting data assessment is to make sure for each individual to reach a steady state during the entire test, and the time it takes to reach their steady state depends on their current physical and physiological characteristics. Participants were appropriately warmed up before the testing, and they were repeatedly encouraged to complete the activity with their “regular” habits and pace.

### 2.6. HR Monitoring

HR was measured simultaneously using a Polar T31 telemetric system (Polar Electro OY, Kempele, Finland). The participants’ HR was monitored minute-by-minute during each activity. Each participant underwent individual calibrations to establish the relationship between HR and energy expenditure in all of the tested activities. The calibration activities were carried out in sequence, with a break to allow recovery of resting HR.

### 2.7. Statistical Analysis

Statistical analysis and algorithm construction were performed using STATA 13.0 (StataCorp LLC, College Station, TX, USA). Data were analyzed for each exercise procedure. This study evaluated the validation between estimated energy expenditure from Cortex MetaMax3B and triaxial acceleration data from an SWA sensor and built energy expenditure algorithms based on seven different activities. Energy expenditure across each activity protocol was analyzed with ANOVA to assess mean differences in different physical activities. Dependent *t*-tests were performed to compare triaxial differences in different activities and within the same physical activity under different speeds/workloads. Statistical significance was defined with *p*-value ≤ 0.05. The gender estimation of energy expenditure was significantly different; therefore, men and women were separated for all analyses. Graphical procedures were used to spot differences in activities with triaxial accelerometer. SenseWear Pro3 Armband™ measures acceleration and deceleration in the three dimensions of space according to ACx (forward accel point), ACy (transverse accel point), and ACz (longitudinal accel point). Additionally, VM (vector magnitude) is calculated as VM = (ACx^2^ + ACy^2^ + ACz^2^)^1/2^.

## 3. Results

### 3.1. Posture and Movement Classification

This study analyzed the validity of the SenseWear Pro3 Armband™ to capture and recognize different modes of activities in a laboratory setting. When the generalized algorithm provided by the manufacturer was applied to the data, the three-axis accelerometer data significantly distinguished these seven activities with ACx, Acy, and ACz reads (see Figure 1).

Tilt sensing is a basic function provided by accelerometers in response to gravity or constant acceleration. Therefore, human postures, such as sitting and lying, can be distinguished according to the signal magnitude of accelerations along sensitive axes [14,15].

### 3.2. Construction and Analysis of Energy Consumption Model of Physical Activity

The data obtained from the triaxial acceleration sensor were imported into STATA 13.0 to match with the data from the gas analyzer, and the non-conforming acceleration data were deleted to ensure that the acceleration value was completely corresponding to the MetaMax 3B data in time. Stepwise regression was used to construct the physical activity energy consumption model. According to relevant studies, the vertical acceleration value of the longitudinal accel (ACz) and VM values have a significant effect on distinguishing different physical activities (*p* = 0.000, *p* < 0.05) [16,17]. The value of seven activities was calculated (as seen in Table 3). No significant difference was found in treadmill walking between slow and fast speeds (*p* = 0.9486, *p* > 0.05).

Pearson correlation analysis was used to analyze the relationship between the BMI value, ACz value, VM value, and energy consumption in the test, and the correlation significance between energy consumption and other test values was statistically calculated. The correlation coefficients between the ACz value, VM value, BMI value, and energy consumption W were 0.59 (*p* = 0.000), 0.76 (*p* = 0.000), and 0.29 (*p* = 0.011), respectively, and all *p*-values were less than 0.05. BMI, ACz, VM, and energy consumption values were significantly correlated. Therefore, it is feasible to construct the energy consumption model with these three variables (as shown in Table 4 and Table 5).

The basic energy consumption algorithm equation is W/min = β0 + β1 × ACz, and can be constructed with additional factors, e.g., sex, BMI, and HR, and the correlation coefficients were 0.46, 0.47, and 0.7, respectively. The accuracy of the algorithm was gradually improved when all of these variables were added to the model.

The comprehensive energy consumption algorithm (based on the ACz) was as follows:W/min = −9.173 + 0.004 × ACz + 1.097 × SEX + 0.227 × BMI + 0.057 × HR (M = 1, F = 0)

The comprehensive energy consumption algorithm (based on the VM axis) was as follows:W/min = −12.27 + 0.008 × VM + 0.953 × SEX + 0.232 × BMI + 0.059 × HR (M = 1, F = 0)

As a result of this study, we developed new proprietary exercise-specific algorithms for common physical activities of Chinese college students. When the exercise-specific algorithms were applied to the data from the SenseWear Pro3 Armband™, the estimate of energy expenditure appeared to be improved. The data were put into the models, and the correlation between the calculated value and the real value was verified by comparing it with the results from Cortex MetaMax3B™. According to the results, the correlation coefficients between the actual energy consumption measured by Cortex MetaMax3B™ and the results calculated by the model were 0.8674 (ACz) and 0.88 (VM), respectively, both of which were greater than 85%, indicating that the algorithms accurately predicted energy consumption for different activities.

Different parameters can be used to build different types of algorithms, and the accuracy of different algorithms varies greatly. Progress has been made to solve these issues, but it is likely that the fundamental challenge is to build an accurate and feasible energy expenditure algorithm with multiple parameters. Direct calorimetry is considered the most accurate method to assess physical activity by measuring gas exchange and interpreting it into energy expenditure. As a result, an algorithm model was built to improve the estimation of energy expenditure when used in combination with parameters such as accelerometry, HR, sex, and BMI.

## 4. Discussion

The current study was conducted to investigate the relationship between energy expenditure and body acceleration during different physical activities. The development of an accurate, reliable, and feasible model to calculate daily energy expenditure in free-living conditions is an important priority for public health researchers. Our study built a correlation regression model based on a triaxial acceleration monitor during seven physical activities, integrating the variables: BMI, sex, and HR. Our correlation coefficient, R, became greater when integrating all three variables than when only using one or two of them, which indicates achieving greater accuracy of the regression model. Biomechanical detection for the movement of the human body and the classification of motion using accelerometer and center of gravity (COG)-based methodologies have been adopted globally. Approaches to the classification of biomechanical movement can be made by threshold-based or statistical-based classification schemes. Threshold-based motion classification takes advantage of well-known knowledge and information about the movements to be classified. Statistical-based motion classification utilizes a supervised machine learning procedure, which associates observations (or features) of movement to possible classifications in terms of the probability of the observation (see Figure 1). Spatial sensing is a basic function of accelerometers, which can respond to the change in the center of gravity or constant acceleration. Therefore, by wearing an accelerometer on the torso or arm, human postures (such as sitting, walking, and running) can be distinguished according to the signal magnitude of accelerations along the measuring axes.

By using the wearable triaxial acceleration sensors, different types of motion can be identified by the three-directional variable, which was obviously distinguished in seven different types of physical activities. In our research, the postures tested can be distinguished by observing different orientations of body segments and the changes in the spatial movement of the body. The sedentary activities (rest and sitting) demonstrated almost the same spatial acceleration image characteristics, whereas the other activities produced distinct spatial acceleration images. Triaxial acceleration signals and discrete wavelet transformations can determine activities in ambulatory movement. Above all these three-dimensional data, vertical acceleration signals are best for distinguishing between different types of motion, as they are characterized by vertical acceleration and frequency peak in the signal spectrum.

Several other studies have investigated the accuracy of posture recognition by accelerometers. Mathie et al. [18] reported a general classification framework consisting of a hierarchical binary tree for classifying postures, e.g., falls, jumps, walks, and other movements, using signals from a wearable triaxial accelerometer. This modular structure also allows modifying the algorithms for each classification under certain conditions or particular purposes. Trunk tilt variation due to a sit–stand postural shift was reported to be measured by integrating the signal from a gyroscope attached to the chest of the examinee [19]. The sit–stand postural shift can be recognized according to the patterns of vertical acceleration from an accelerometer at the waist [20]. Although Yang et al. [21] used a simplified scheme with a tilt threshold to distinguish standing and sitting, a single-accelerometer approach has difficulty in distinguishing between standing and sitting as both are upright postures.

Information from the sensors together with sex, BMI, and HR was integrated into proprietary algorithms to estimate energy expenditure (EE). The significant correlations between energy expenditure and accelerometer readings are found in the laboratory and under free-living conditions, and the relationship between these parameters using triaxial accelerometers varies between different types of physical activities. From the results of our research based on Chinese college students, we developed new proprietary exercise-specific algorithms for energy expenditure prediction. When the exercise-specific algorithms were applied to the data from the control group and energy cost from Cortex MetaMax3B^TM^, the estimate of energy expenditure appeared to be improved. To facilitate comparison, we classified the seven different activities into the walking and running group and the non-walking and running group (rest, sitting, cycle ergometer, pushup, and stair walking). An energy consumption correlation analysis was carried out for the two groups, as illustrated in Figure 2. There was a linear correlation between total energy expenditure examined via the Cortex MetaMax3B^TM^ and total energy expenditure estimated using indirect calorimetry in this study.

The distribution of the scatter plot shows that the values calculated by the energy consumption algorithm based on ACz and VM have a good fit with the actual measured data from Cortex MetaMax3B^TM^. As can be seen from the scatter plot, the energy consumption of the walking and running group is higher than that of the non-walking and running group. The energy consumption algorithm based on ACz shows a higher difference between the two groups, whereas the energy consumption algorithm based on VM shows a small difference between the two groups. The reason may be that the VM value is a composite value of three-dimensional acceleration, which can reduce the difference when calculating the energy consumption of different types of physical activities. Relevant studies have shown that there is a high correlation between the three-dimensional space axis and PAEE, and the linear algorithm is easy to calculate, so most researchers construct linear equations and use ACz/VM as the independent variable to predict the energy consumption of physical activity [22]. The results show that the correlation between the energy consumption algorithm and the IC method is between 0.50 and 0.90. The classical energy consumption prediction formula, the Freedson formula, is based on a treadmill with three different speeds: 4.8 km/h, 6.4 km/h, and 9.7 km/h, and the R^2^ of the energy consumption algorithm is 0.82 [23]. Other researchers also took walking and running as normal daily activities and established multiple energy consumption models with ACz or VM as independent variables, gradually improving the comprehensiveness of the test and the accuracy of the prediction [24].

Different accelerometers have different validity in adult physical activity studies, and the SenseWear Pro3 Armband™ and some other accelerometers have been used during exercise to assess energy expenditure. For example, the research found that the SenseWear Pro3 underestimated energy expenditure, particularly for monitoring high-intensity exercise. However, there is currently no cure for improving the underestimation; therefore, the underestimation of energy expenditure continues to be a problem of many accelerometer-based physical activity monitors that are currently available. In a study comparing a triaxial accelerometer to indirect calorimetry, Jakicic et al. [25] reported that the accelerometer under-evaluated energy expenditure by a total of 30–50 kcal for 30 min of walking, 87–89 kcal for 20 min of cycling, and 44–51 kcal for 20 min of stair stepping. In another study of Jakicic et al. [26], the difference between indirect calorimetry and the SenseWear Pro3 Armband™ was studied by using similar exercise protocols; the total energy expenditure was also underrated. Jakicic found that the SenseWear Pro3 Armband™ is able to give an accurate estimate of energy expenditure of exercises mainly involving the upper extremities [26]. These results suggest that the SenseWear Pro3Armband™ may provide a more accurate estimate of energy expenditure using exercise-specific algorithms. Hustvedt et al. [27] evaluated the accuracy of the ActiReg (a three-dimensional accelerometer) alone and in combination with an HR monitor. The mean TEE (total energy expenditure) measured by the ActiReg was not different from DLW (doubly labeled water) (*p* = 0.45). Bland–Altman plots showed that the ActiReg underestimated TEE at high-intensity exercise, and the underestimation of TEE was corrected by using an HR monitor. Plasqui et al. [28] evaluated another three-dimensional monitor called the Tracmor. The participants’ age, body mass, and height were shown to explain 64% of the variation in DLW-measured TEE, and by adding Tracmor activity counts to the model, there was an increase in explained variation of 19% (total R^2^ = 0.83). Our study suggests that when exercise-specific algorithms are used in combination with triaxial acceleration, sex, BMI, and HR, this results in providing a more accurate estimate of energy expenditure, indicating that the body shifts and other physiological sensors provide useful information to improve estimates of energy expenditure.

Despite the promising results obtained from this study, there are limitations that need to be discussed. In the current study, the researchers focused on the accuracy of exercise-specific algorithms and many different factors, such as age, gender, and activity forms, which need to be considered to estimate energy expenditure. The exercise-specific algorithms from our study are based on a laboratory study, and may not be as accurate under free-living conditions. This study is based on Chinese college students, and the participants are relatively young adults with normal body weights. It is unclear whether our findings hold true for individuals of different ages, weights, or levels of physical fitness.

## 5. Conclusions

In summary, by the use of a triaxial acceleration sensor, body postures of seven physical activities were distinguished, and the algorithms showed promise for accurately measuring energy expenditure. On the one hand, the accuracy of the algorithms was improved by the additional three parameters (sex, BMI, and HR). On the other hand, the manufacturers of some accelerometers have already written the algorithms for their products, but they still need to improve the accuracy of the original data collected during each of the exercises to estimate energy expenditure. In future research, the accuracy of the model can be continuously improved by increasing the number of subjects. Meanwhile, the physical activities for measuring energy consumption can be more diversified and include activities such as outdoor sports, various ball games, and competitive and confrontational sports, such as tennis, table tennis, badminton, swimming, basketball, and football. In the process of constructing the energy consumption algorithm, different analysis methods can be added, such as the piecewise model, neural network model, etc., to enhance the accuracy of the actual prediction of energy expenditure.

## Figures and Tables

**Figure 1 ijerph-20-05184-f001:**
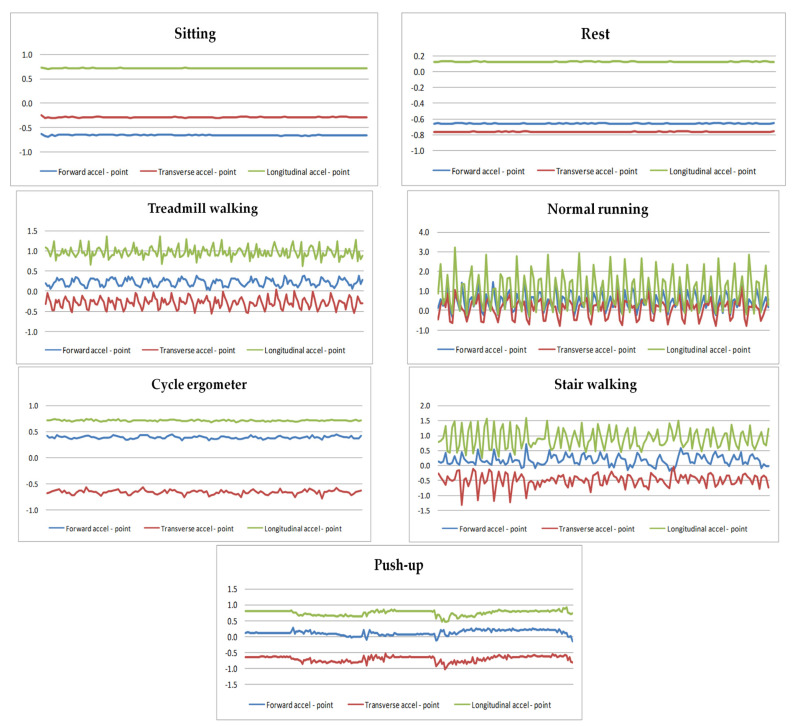
Standardized data obtained from SenseWear Pro3 Armband™, three different types of curves marked the seven tested activities, triaxial acceleration data indicated with forward accel-point, transverse accel point, and longitudinal accel point.

**Figure 2 ijerph-20-05184-f002:**
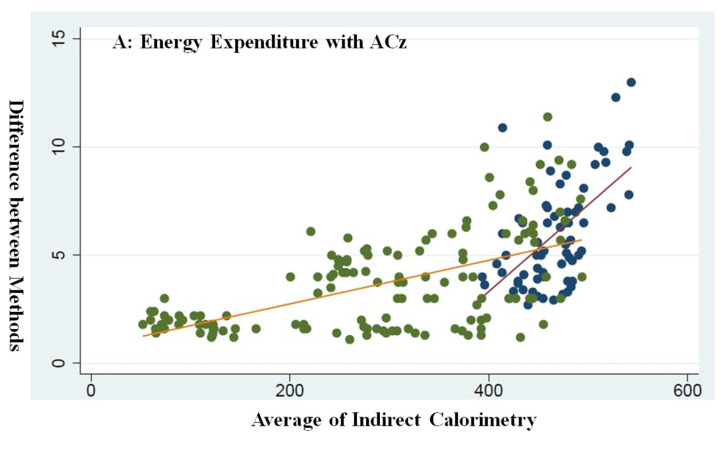
(**A**) Scatter plot showing the energy expenditure assessed by energy consumption algorithm (based on the ACz) for control subjects across seven activities (*n* = 20; 10 males and 10 females). (**B**) Plot showing the energy expenditure assessed by energy consumption algorithm (based on the VM) for control subjects across seven activities (*n* = 20; 10 males and 10 females). Blue spot indicates walking and running group with red solid line as mean value, and green spot indicates the non-walking and running group with orange solid line as mean value.

**Table 1 ijerph-20-05184-t001:** Participant characteristics (*n* = 100, mean ± SD).

	EXPERIMENTAL GROUP		CONTROL GROUP	
	Male (*n* = 40)	Female (*n* = 40)	Male (*n* = 10)	Female (*n* = 10)
Age (Year)	20 ± 3.1	19 ± 1.7	23 ± 1.7	21 ± 1.1
Height (cm)	174.8 ± 6.6	164.5 ± 5.6	174.7 ± 5.6	162.6 ± 5.4
Weight (kg)	66.6 ± 11.3	57.4 ± 9.1	64.0 ± 8.1	55.2 ± 7.3
Body Mass Index (kg/m^2^)	21.7 ± 3.5	21.4 ± 3.3	21.0 ± 2.4	20.8 ± 2.3

**Table 2 ijerph-20-05184-t002:** Description of Exercise Protocols.

Activity	Description	Duration (min)	Speed (km/h)
Rest	Lying down on bed quietly, being awake, breathing softly with eyes closed, and listening to soft music in headphones	10	
Sitting	Sitting quietly, playing video game	10	
Treadmill walking	Walking on a treadmill at different speeds, defined as slow (3.2 km/h) and fast (4.8 km/h)	5	3.2
		5	4.8
Normal running	Running on a treadmill (8.1 km/h)	5	8.1
Stair walking	Walking up and down stairs with 35 steps/min	5	
Cycle ergometer	60 RPM with no external load	5	
Push-up	Push up and down at a pace of 25/min	5	

**Table 3 ijerph-20-05184-t003:** Acceleration values of seven activities (mean ± SD).

Activity	ACz	VM
Rest	94.37 ± 17.22	474.86 ± 15.92
Treadmill walking	412.38 ± 37.36	497.61 ± 14.71
Sitting	311.64 ± 62.06	450.47 ± 18.19
Cycle ergometer	266.43 ± 55.41	497.62 ± 50.17
Normal running	573.61 ± 98.38	647.57 ± 85.25
Push-up	435.92 ± 103.26	488.04 ± 37.60
Stair walking	486.16 ± 110.37	501.12 ± 107.13

ACz—longitudinal accel point generated in counts/min; VM—vector magnitude generated in counts/min.

**Table 4 ijerph-20-05184-t004:** Energy consumption model (based on ACz).

	Model 1	Model 2	Model 3	Model 4
Subject	EE (W/min)	EE (W/min)	EE (W/min)	EE (W/min)
ACz (per min)	0.010 **	0.012 **	0.011 **	0.004 **
Sex		1.320 **	1.095 **	1.097 **
BMI			0.132 *	0.227 **
HR (per min)				0.057 **
Constant term	0.290	−0.483	−3.09 *	−9.173 **
R^2^	0.403 **	0.464 **	0.470 **	0.701 **

** *p* < 0.01, * *p* < 0.05; ACz—longitudinal accel point generated in counts/min.

**Table 5 ijerph-20-05184-t005:** Energy consumption model (based on VM).

	Model 1	Model 2	Model 3	Model 4
Subject	EE (W/min)	EE (W/min)	EE (W/min)	EE (W/min)
VM (per min)	0.022 **	0.022 **	0.022 **	0.008 **
Sex		0.914 **	0.711 *	0.953 **
BMI			0.147 *	0.232 **
HR (per min)				0.059 **
Constant term	−7.112 *	−7.345 *	−10.126 **	−12.270 **
R^2^	0.385 **	0.416 **	0.431 **	0.709 **

** *p* < 0.01, * *p* < 0.05; VM—vector magnitude generated in counts/min.

## Data Availability

Data are available on request due to privacy and ethical restrictions.

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
