# Peer review of "Assessing Physical Activity Levels among Chinese College Students by BMI, HR, and Multi-Sensor Activity Monitors"

_ijerph, 2023, doi:10.3390/ijerph20065184_

Round 1

Reviewer 1 Report

This is an interesting study examining the use of multi-sensor physical activity monitors to measure energy expenditure of various physical activity levels of Chinese collegiate students. According to the results, the model of predicting physical activity energy consumption based on multi-sensor physical activity monitors, BMI, and HR demonstrated high accuracy, and thus it can be applied to daily physical activity monitoring processes.

Authors may consider some issues highlighted below to increase clarity and improve the quality of their manuscript.

Introduction is very short and should be enriched with more information regarding the objective of these study and the other measures used for monitoring energy expenditure. For example, the Doubly labeled water (DLW) method is presented, without any information about it. This makes difficult for unfamiliar readers to understand and to compare authors’ method with the previous ones. Thus, authors should further elaborate this part of introduction referring to measures of monitoring energy expenditure. This information should be also accompanied with related empirical evidence.

Authors, should also elaborate their arguments regarding the rational of their study. Why actually is needed a new method of monitoring physical activity energy expenditure? What are the advantages of this new method compared to the previous ones? Most importantly, are there any practical advantages of the proposed method compared to the previous ones?

Another issue for this study is that authors stated that this study had a control group. However, it is not clear what was the nature of this group and what actually control group students did in this study. Authors should clarify this issue providing respective information.

line 91: please do not start a new paragraph with a number

line 94: please clarify how students “were aware of the procedures and purposes of the study”

line 97: you may consider using the term “experimental group” instead of “sample”

lines 105-108: this information should not be included in the measures section but rather in a section describing the procedures of the study

Table 2: what do you mean with the “Push up and down at a pace of 25”. In this activity, students had to perform push-ups for 5 minutes? What happened if a student could not perform push-ups for all this time?

Similarly, authors should provide more information regarding students’ performance in all activities involved and if all students performed as they were required.

Line 257: authors may reconsider using the term “daily physical activities”. Some of the test included cannot be considered as such activities or other are laboratory-like physical activities

Line 259: please explain what do you mean with the phrase “free-living circumstances”

This study included a laboratory experiment involving specific tests. Although authors mentioned this as a limitation, they may consider elaborating more this issue and discussing potential variations in their results if other types of tests were used.

Moreover, although the results of this study came from a laboratory experiment, authors may discuss some practical implications of their study.

Reviewer 2 Report

This on the whole is a well thought out study and well written paper. Please do make the following amendments though:

Abstract:

1.       Body Mass Index (BMI) on line 13.

2.       Results – what were the pre and post values (lines 21-23) what was the p-value as you stated there were significant differences?

Introduction:

1.       Lines 85 -88. Elaborate on what you mean by ‘additional relevant parameters’.

   Methods:

1)      Line 91 – 18-25 years.

2)      Is a split of 8- sample and 20 control a fair split – justify the decision.

3)      Name of digital scale? Put in brackets.

4)      What height measurer – put in brackets.

5)      Table 2 – what was the intensity on the cycle ergometer, did it differ persona, or same?

6)      Push ups – 25 per what?? A minute?

7)      Lines 133-136 – what is the reason for sampling frequency every 0.1 sec and peak acceleration and MAD every 6 sec?

8)      Line 141 – “breath-by-breath, which has been…” add in ‘which’.

9)      Line 145 say what O2 and CO2 are in full and abbreviate the first time you mention them.

1)   Line 147 – millilitres

1)   Line 170 – “with an ANOVA”

1)   Line 177 – state what VM is.

1)   Lines 176-178 – these don’t flow as well as previous sentences, where you write ‘triaxial’ this seems to finish suddenly, is ‘deprived’ the correct word?

   Results:

1.       Line 187 should it be ‘drawn’?

2.       Line 203 ‘walking’

3.       Table 3 - a key is needed below for ACz and VM

4.       Table 4 – units and a key are needed.

5.       Table 5 – units and a key are needed.

Discussion:

1.       I would like to see what novel findings you have discovered in this research within the first paragraph.

2.       Line 271 – accelerometers

3.       Line 354-256 – does not flow as well, is this relating to your study or Jakicic’s make it clear/re-word.

4.       Define TEE

5.       Define DLW

6.       Line 374 – add ‘which’… “forms, which need to be..”
